# Epidemiological and Antimicrobial Resistance Trends in Bacterial Keratitis: A Hospital-Based 10-Year Study (2014–2024)

**DOI:** 10.3390/microorganisms13030670

**Published:** 2025-03-17

**Authors:** Qingquan Shi, Deshuo Mao, Zijun Zhang, Ahyan Ilman Qudsi, Mingda Wei, Zhen Cheng, Yang Zhang, Zhiqun Wang, Kexin Chen, Xizhan Xu, Xinxin Lu, Qingfeng Liang

**Affiliations:** Beijing Institute of Ophthalmology, Beijing Tongren Eye Center, Beijing Tongren Hospital, Capital Medical University, Beijing 100005, China; qingquan@mail.ccmu.edu.cn (Q.S.); 2114001@mail.ccmu.edu.cn (D.M.); shenyu@ccmu.edu.cn (Z.Z.); ahyanqudsi@mail.ccmu.edu.cn (A.I.Q.); weimingda@mail.ccmu.edu.cn (M.W.); 1901156@mail.ccmu.edu.cn (Z.C.); biozy1@126.com (Y.Z.); eyewzq@163.com (Z.W.); ckxwl1234@hotmail.com (K.C.); xuxz0924@mail.ccmu.edu.cn (X.X.); luxinxin2009@126.com (X.L.)

**Keywords:** bacterial keratitis, antimicrobial resistance, fluoroquinolone

## Abstract

Bacterial keratitis (BK) is a severe ocular infection that can lead to vision loss, with antimicrobial resistance (AMR) posing a growing challenge. This study retrospectively analyzed 1071 bacterial isolates from corneal infections over a 10-year period (2014–2024) at a tertiary ophthalmic center in Beijing, categorizing them into three distinct phases: pre-COVID-19, during COVID-19, and post-COVID-19. The results indicated significant changes in pathogen distribution, including a marked decrease in Gram-positive cocci (from 69.8% pre-COVID-19 to 49.3% in post-COVID-19, *p* < 0.001), particularly in *Staphylococcus epidermidis*. In contrast, Gram-positive bacilli, particularly *Corynebacterium* spp., increased from 4.2% to 16.1% (*p* < 0.001). The susceptibility to gatifloxacin, moxifloxacin, and ciprofloxacin significantly declined in both Gram-positive cocci and bacilli during the COVID-19 period (all *p* < 0.01). Gatifloxacin resistance in *Staphylococcus* rose from pre-COVID-19 (15.2%) to COVID-19 (32.7%), remaining high post-COVID-19 (29.7%). A similar trend was observed in *Streptococcus* and *Corynebacterium*, where resistance rose sharply from 12.0% and 22.2% pre-COVID-19 to 42.9% during COVID-19, and remained elevated at 40.0% and 46.4% post-COVID-19, respectively (*p* < 0.01). These findings emphasize the rapid rise of fluoroquinolone resistance in several bacterial groups, underscoring the urgent need for continuous surveillance and improved antimicrobial stewardship to enhance treatment outcomes.

## 1. Introduction

Bacterial keratitis (BK) is a serious condition that can lead to rapid corneal destruction and permanent vision loss. The increasing problem of antimicrobial resistance (AMR) further complicates treatment [1,2,3]. The microbial composition of corneal infections is influenced by various factors, including antibiotic use, environmental changes, and medical practices [4,5]. The COVID-19 pandemic triggered unprecedented global changes, affecting healthcare accessibility, hygiene practices, and antibiotic prescription patterns, which may have contributed to shifts in bacterial epidemiology [6,7].

Several studies have reported changes in the prevalence and AMR patterns of bacterial pathogens during and after the COVID-19 pandemic [8,9,10]. In the field of ophthalmology, emerging studies from the UK, Portugal, and Mexico suggest that the COVID-19 pandemic may have altered the microbiological landscape of infectious keratitis [11,12,13]. For instance, in the UK, despite comparable visual outcomes between pre- and post-lockdown, the proposition of poly-microbial cultures significantly decreased [12]. Concurrently, Portuguese data revealed a rise in culture positivity rates, driven by an increased prevalence of *Pseudomonas aeruginosa* and *Corynebacterium*, contrasting with the finding in Mexico [11,13]. However, studies specifically investigating antibiotic susceptibility patterns in BK remain scarce, leaving a gap in understanding how the pandemic may have influenced resistance trends. Identifying these trends is essential for developing empirical treatment protocols and antimicrobial stewardship strategies tailored to broader populations.

To systematically assess the pandemic’s impact, we divided the study into three phases based on distinct epidemiological and public health milestones: (1) pre-COVID-19 (2014–2019), representing baseline microbial patterns before SARS-CoV-2 emergence; (2) during COVID-19 (2020–2022), covering the peak pandemic period with widespread lockdowns, altered healthcare access, and increased antibiotic use for respiratory infections; and (3) post-COVID-19 (2023–2024), reflecting transitional phases where pandemic-related restrictions were lifted but residual effects on microbial ecology persisted [14]. This stratification enables the identification of both immediate and prolonged shifts in pathogen dynamics and resistance profiles.

This study analyzes bacterial isolates from corneal infections between 2014 and 2024, covering three distinct periods: pre-COVID-19, during COVID-19, and post-COVID-19. We focus on the distribution of Gram-positive cocci, Gram-positive bacilli, and Gram-negative bacilli, and analyze their epidemiological trends. Additionally, we also evaluate the antimicrobial susceptibility patterns of the bacteria to identify changes in resistance. With a systematic analysis of the microbiological characteristics of BK before, during, and after the pandemic, this study aims to deepen the understanding of the disease’s epidemiological trends and provide insights for future treatment strategies.

## 2. Materials and Methods

### 2.1. Strains Source

This retrospective study investigated 1071 bacterial strains isolated from corneal samples of 970 BK patients at Beijing Tongren Hospital between 1 January 2014 and 1 July 2024. As the largest tertiary ophthalmic specialty hospital and regional referral center in northern China, Beijing Tongren Hospital adheres strictly to international standards in corneal disease diagnosis and treatment. All enrolled patients met both clinical and microbiological diagnostic criteria for BK, including slit-lamp findings of corneal epithelial defects with gray-white stromal infiltrates (≥1 mm in diameter), hypopyon, and conjunctival injection. To exclude non-bacterial pathogens, cases with co-infections involving fungi, viruses, or parasites were systematically excluded. The study was conducted in accordance with the Declaration of Helsinki and approved by the Ethics Committee of Beijing Tongren Hospital (TRECKY2021-024. F1).

### 2.2. Strain Isolation

Microbiological investigations were performed on patients suspected of corneal infections. To confirm bacterial etiology, corneal scrapings were aseptically collected from the ulcer margins under topical anesthesia using a sterile surgical blade, avoiding contamination from adjacent ocular surfaces. The investigation included microscopic examination of corneal scrapings, Gram staining, followed by microbial culture and antimicrobial susceptibility testing. For Gram staining, a thin smear of the specimen was prepared on a clean glass slide and heat-fixed. The slide was then stained with crystal violet for 1 min, rinsed with distilled water, and treated with Gram’s iodine solution for 1 min. After a brief rinse, the slide was decolorized with acetone-alcohol for 30 s and rinsed immediately. A counterstain with safranin was applied for 1 min, followed by a final rinse and air-drying. The stained slide was examined under a microscope to detect and differentiate bacterial cells based on cell wall properties.

For culture, the remaining specimen was inoculated onto both chocolate agar and blood agar plates using aseptic techniques. The inoculated plates were incubated at 35 °C (21% O_2_) in an aerobic environment for 18–24 h. After incubation, plates with bacterial growth were assessed, and representative colonies were selected for identification. Selected colonies were processed using the MALDI-TOF-MS system (Bruker Corporation, Bremen, Germany) to identify the bacterial species responsible for the infection, with a species-level identification threshold set at a log-score ≥2.0. All diagnostic assessments were conducted by experienced ophthalmologists and trained laboratory staff. All procedures were performed using calibrated instruments and standard reagents, with results documented in the laboratory information management system.

### 2.3. Antibiotic Susceptibility Testing

Antimicrobial susceptibility testing was performed using the Kirby–Bauer disk diffusion method, and the interpretation of results followed the Clinical and Laboratory Standards Institute (CLSI) guidelines [15]. The antibiotic panel (rifampin, moxifloxacin, ciprofloxacin, gatifloxacin, amikacin, gentamicin, ceftazidime, ofloxacin, tobramycin) was selected based on empirical treatment guidelines for BK and local antibiotic availability. We referenced the CLSI guidelines to determine susceptibility thresholds, with the detailed breakpoints for each antibiotic documented in Appendix A.

Isolates were standardized to a 0.5 McFarland turbidity standard, achieving a target inoculum density of approximately 1.5 × 10^8^ CFU/mL. The standardized bacterial suspension was evenly spread over Mueller–Hinton agar plates using a sterile swab to form a uniform bacterial lawn. After the inoculum settled briefly, antibiotic-impregnated disks were carefully placed on the agar surface using sterile tweezers. The plates were then incubated at 35 °C (21% O_2_) in an aerobic environment for 18–24 h. After incubation, the diameters of the inhibition zones around each antibiotic disk were measured to the nearest millimeter. Interpretations were aligned with CLSI criteria, classifying isolates as susceptible (S), intermediate (I), or resistant (R) according to species-specific zone diameter thresholds. Additionally, quality control strains (e.g., *Escherichia coli* ATCC 25922) were integrated monthly to validate the accuracy of antimicrobial susceptibility testing for uniformity in susceptibility testing.

### 2.4. Statistical Analysis

We applied R 4.4.2 (R Foundation for Statistical Computing, Vienna, Austria) to perform statistical analyses. Descriptive statistics were calculated, and data were presented in graphical form. Central tendency and dispersion measures were used to describe continuous variables, while categorical variables were noted as percentages. Chi-square tests or Fisher’s exact tests were used in the comparison of changes in strain counts and the numbers of susceptible/intermediate/resistant strains across different periods (pre-COVID-19, COVID-19, and post-COVID-19). Bonferroni correction was used to interpret multiple comparisons. Statistical significance was defined at *p* ≤ 0.05.

## 3. Results

### 3.1. Bacterial Profile Shifts Across the COVID-19 Period

The results of bacterial isolates from the cornea during the pre-COVID-19 (2014–2019), COVID-19 (2020–2022), and post-COVID-19 (2023–2024) periods are summarized in Table 1, revealing significant shifts in bacterial profiles that may be linked to the pandemic. Gram-positive cocci, the most predominant group (62.9%, 674/1071), exhibited a significant decline in prevalence from 69.8% (460/659) in the pre-COVID-19 period to 54.7% (110/201) during COVID-19 and further to 49.3% (104/211) in the post-COVID-19 period (*p* < 0.001). This decline was especially noticeable in *Staphylococcus epidermidis*, which fell from 31.1% (205/659) pre-COVID-19 to 17.5% (37/211) post-COVID-19 (*p* < 0.001). In contrast, *Staphylococcus aureus* remained stable at approximately 5% across all periods (*p* = 0.992).

Gram-positive bacilli exhibited a significant increase during and after the pandemic, rising from 8.6% (57/659) pre-COVID-19 to 18.4% (37/201) during COVID-19 and further to 21.3% (45/211) post-COVID-19 (*p* < 0.001). This trend was largely driven by *Corynebacterium* spp., which increased from 4.2% (28/659) pre-COVID-19 to 16.1% (34/211) post-COVID-19 (*p* < 0.001). Similarly, *Mycobacterium* spp. showed a notable rise during the COVID-19 period (3.0%, 6/201); however, its prevalence decreased slightly post-COVID-19 (0.5%, 1/211) (*p* = 0.001). Gram-negative bacilli, accounting for 23.5% (252/1071) of isolates, showed no significant change in overall prevalence across the three periods (*p* = 0.057). *Pseudomonas* spp., the most common Gram-negative bacillus (7.8%, 84/1071), remained stable throughout (*p* = 0.681). Among the less common bacterial isolates, *Abiotrophia* spp. accounted for 1.0% (11/1071) of the total isolates, with minimal variation across the three periods (*p* = 0.649). *Nocardia* spp., which represented 1.1% (12/1071) of isolates, showed a slight increase during COVID-19 (2.0%, 4/201) compared to pre-COVID-19 (0.6%, 4/659) and remained stable post-COVID-19 (1.9%, 4/211) (*p* = 0.130). *Haemophilus* spp. and *Enterobacter* spp. were rare, with *Haemophilus* spp. showing a slight increase post-COVID-19 (1.4%, 3/211) compared to pre-COVID-19 (0.6%, 4/659) (*p* = 0.441), while *Enterobacter* spp. was absent post-COVID-19 (*p* = 0.213). Gram-negative cocci were rare (0.6%, 6/1071), with no significant changes observed (*p* = 0.130).

### 3.2. Overall Antibiotic Susceptibility of Bacteria

The overall susceptibility rates of bacterial isolates from the cornea to various antibiotics, categorized into Gram-positive cocci, Gram-positive bacilli, and Gram-negative bacilli, are shown in Figure 1. Gram-positive cocci were the most predominant group, with *Staphylococcus epidermidis* and *Staphylococcus aureus* being the most clinically significant. Overall, these bacteria showed high susceptibility to rifampicin (e.g., *S. epidermidis* at 0.94 and *S. aureus* at 0.96). Both *S. epidermidis* and *S. aureus* exhibited higher susceptibility to gentamicin (0.56 and 0.70) than *Streptococcus* spp. (0.24, 0.24 and 0.38). Within the *Streptococcus* spp., different species exhibit notable variations in antibiotic susceptibility. *S. pneumoniae* and *S. mitis* show low susceptibility to tobramycin (0.02 and 0.13) compared to *S. oralis* (0.95). Additionally, *S. mitis* and *S. oralis* are highly susceptible to ceftazidime (0.94 and 0.78), while *S. pneumoniae* shows lower susceptibility to the same antibiotic (0.25).

*Corynebacterium* spp. and *Mycobacterium* spp. were the most prominent among Gram-positive bacilli. Both *Corynebacterium* spp. and *Mycobacterium* spp. exhibited high susceptibility to rifampicin (range: 0.92–1.00). However, *Mycobacterium* spp. exhibited lower susceptibility to gatifloxacin (0.33) compared to *Corynebacterium* spp. (range: 0.54–0.57). Gram-negative bacilli, particularly *Pseudomonas aeruginosa*, displayed high susceptibility to various antibiotics, including amikacin (0.88), ciprofloxacin (0.89), ofloxacin (0.88), gatifloxacin (0.88), and ceftazidime (0.96). Moreover, quinolones, such as ciprofloxacin and ofloxacin, are highly effective against Gram-negative bacilli (all susceptibility rates above 0.50), while some aminoglycosides, such as gentamicin, exhibited greater variability in their efficacy (susceptibility rates range from 0 to 1.00).

### 3.3. Trends in Bacterial AMR Across the COVID-19 Pandemic

The AMR profiles of Gram-positive cocci, Gram-positive bacilli, and Gram-negative bacilli exhibited distinct trends across the pre-COVID-19, COVID-19, and post-COVID-19 periods (Figure 2). Among Gram-positive cocci, susceptibility to quinolone antibiotics (ciprofloxacin, gatifloxacin, and moxifloxacin) significantly declined during the COVID-19 period, with resistance rates increasing consistently over time, suggesting a potential link between increased antibiotic use and selection pressure (all *p* < 0.001). Similarly, Gram-positive bacilli showed a progressive reduction in susceptibility to quinolones, including ciprofloxacin, gatifloxacin, moxifloxacin, and ofloxacin, during the COVID-19 period (all *p* < 0.01), suggesting a potential decline in the efficacy of these antibiotics. These findings indicate that the pandemic may have contributed to shifts in patterns in Gram-positive bacteria, potentially due to alteration in antibiotic usage or selective pressures in healthcare settings.

In contrast, the resistance patterns of Gram-negative bacilli remained relatively stable; however, susceptibility to ofloxacin and tobramycin significantly decreased in the post-COVID-19 period (*p* < 0.05), raising concerns about the potential long-term effects of antimicrobial exposure during the pandemic on Gram-negative bacterial populations. The increasing resistance to aminoglycosides and quinolones, particularly in both Gram-positive and Gram-negative bacterial groups, highlights emerging antimicrobial resistance risks.

Gatifloxacin is a commonly used fluoroquinolone antibiotic in ophthalmology. Its resistance patterns have shown distinct trends among *Staphylococcus*, *Streptococcus*, *Pseudomonas*, and *Corynebacterium* over different time periods (Figure 3). In *Staphylococcus*, the proportion of resistant strains increased significantly from 15.2% in the pre-COVID-19 period to 32.7% during COVID-19 (*p* < 0.01). Similarly, *Streptococcus* demonstrated a significant rise in resistance, from 12.0% before COVID-19 to 42.9% during COVID-19, which remained elevated at 40.0% in the post-COVID-19 period (*p* < 0.001). In contrast, *Pseudomonas* exhibited a relatively stable resistance pattern, with resistance rates fluctuating only slightly from 11.1% in the pre-COVID-19 period to 6.2% in the post-COVID-19 period. However, *Corynebacterium* showed a marked shift, with resistance increasing from 22.2% in the pre-COVID-19 period to 42.9% during the COVID-19 period and further rising to 46.4% in the post-COVID-19 period (*p* < 0.01). The overall trend indicates that while resistance to gatifloxacin significantly increased in certain bacterial groups (e.g., *Staphylococcus*, *Streptococcus*, and *Corynebacterium*), others (e.g., *Pseudomonas*) remained relatively stable. These findings underscore the need for continuous surveillance and optimized antibiotic stewardship strategies to mitigate the long-term impact of AMR in the post-COVID-19 period. This can be achieved through enhanced monitoring programs, targeted interventions to reduce unnecessary antibiotic use, and the development of novel therapeutic approaches to counteract resistant bacterial strains.

## 4. Discussion

The COVID-19 pandemic has profoundly altered global healthcare practices, with cascading effects on infectious disease epidemiology and AMR [9,16]. This study, spanning BK cases from 2014 to 2024 at a major ophthalmic center in northern China, provides critical insights into the pandemic’s influence on corneal pathogen distribution and antibiotic susceptibility. Our findings reveal significant shifts in bacterial prevalence and resistance patterns, underscoring the dynamic interplay between public health interventions, antibiotic stewardship, and microbial adaptation.

Consistent with reports from a tertiary ophthalmic center in Portugal [11], our study conducted a subclassification analysis and found that the proportion of Gram-positive cocci decreased, while the proportion of Gram-negative cocci increased, aligning with their findings. The decline in Gram-positive cocci, particularly *Staphylococcus epidermidis*, during and after the pandemic correlates with reports of reduced community-acquired infections attributed to improved hygiene measures (e.g., mask-wearing and hand sanitization) during COVID-19 [16,17,18]. These practices likely limited ocular surface colonization by commensal skin flora [19]. Conversely, the marked rise in Gram-positive bacilli, notably *Corynebacterium* spp., suggests a niche shift toward opportunistic pathogens associated with disruptions in ocular microbiota equilibrium, aligning with previous studies [20,21]. Increased antibiotic use during the pandemic might have selectively suppressed competing flora, potentially facilitating the proliferation of *Corynebacterium*, a known biofilm-forming organism [22]. Similarly, the transient surge in *Mycobacterium* spp. during the pandemic could reflect altered healthcare-seeking patterns or delayed diagnoses due to restricted hospital access, enabling colonization by atypical pathogens [23,24,25,26]. Among the less common bacterial isolates, temporal variations were observed, albeit with limited statistical significance. For example, *Nocardia* spp. exhibited a slight increase during the COVID-19 period compared to the pre-COVID-19 period and remained relatively stable post-COVID-19. It raises the possibility that changes in clinical practices, such as increased corticosteroid use and immunosuppressive treatments during the pandemic, may have contributed to a more favorable environment for opportunistic infections like *Nocardia* spp. [27,28].

The discrepancy between previous studies highlights geographical variation: research from Portugal suggests an increase in *Pseudomonas aeruginosa*, while studies from Mexico report a decline [11,13]. Intriguingly, our findings indicate relative stability in its prevalence. This stability may underscore the unique microenvironment of the cornea, where the inherent resilience and resistance of *Pseudomonas aeruginosa* to environmental stressors serve to buffer it against short-term epidemiological fluctuations [29].

Among the major Gram-positive cocci, *Staphylococcus epidermidis* and *Staphylococcus aureus* both exhibited extremely high susceptibility to rifampicin (0.94 and 0.96, respectively), which supports the continued use of this antibiotic in treating infections caused by these pathogens. According to a study analyzing the profiles and evolution of *Staphylococcus aureus* in pediatric intensive care units (PICUs) across 17 hospitals in China from 2016 to 2022, even among methicillin-resistant *Staphylococcus aureus* (MRSA) isolates, resistance to rifampicin was remarkably low. In this study, out of 234 MRSA strains, only 8 isolates (3.4%) were resistant to rifampicin [30]. This discrepancy might be attributed to inherent resistance mechanisms or prior antibiotic exposure, underscoring the need for species-specific antimicrobial strategies. Moreover, this evidence suggests that rifampicin can be considered a promising option in the treatment of MRSA infections, contributing to improved outcomes in pediatric intensive care settings.

Within the *Streptococcus* group, notable differences were observed. *Streptococcus pneumoniae* and *Streptococcus mitis* exhibited very low susceptibility to tobramycin (0.02 and 0.13, respectively), whereas *Streptococcus oralis* showed very high susceptibility (0.95). Additionally, although *S. mitis* and *S. oralis* were highly susceptible to ceftazidime (0.94 and 0.78, respectively), *S. pneumoniae* demonstrated a much lower susceptibility (0.25). This trend is consistent with previous research, which has also reported significant interspecies differences within the *Streptococcus* group regarding antibiotic susceptibility [31]. These variations suggest that the effectiveness of both tobramycin and ceftazidime can vary greatly among different *Streptococcal* species, which should be carefully considered when devising empirical treatment plans.

For Gram-positive bacilli, both *Corynebacterium* and *Mycobacterium* species displayed high susceptibility to rifampicin. However, *Mycobacterium* spp. showed significantly lower susceptibility to gatifloxacin (0.33) compared to *Corynebacterium* spp. (0.54–0.57), possibly reflecting differences in resistance mechanisms [32,33]. Overall, these findings highlight the importance of tailoring antibiotic therapy to specific pathogens to optimize treatment strategies for corneal infections.

The reduced susceptibility of Gram-positive bacteria to fluoroquinolones during the COVID-19 pandemic arises from a multifactorial interplay of antibiotic misuse, selective pressure, and bacterial adaptability [9]. The widespread use of systemic fluoroquinolones for respiratory infections—despite their ineffectiveness against SARS-CoV-2—may have contributed to increased elective pressure on ocular pathogens [34]. Among *Staphylococcus*, *Streptococcus*, and *Corynebacterium* spp., the rise in gatifloxacin resistance was significant (e.g., *Staphylococcus*: 15.2% to 32.7%; *Streptococcus*: 12.0% to 42.9%; *Corynebacterium*: 22.2% to 42.9%). In *Staphylococcus* spp., fluoroquinolone resistance primarily stems from mutations in target genes, such as gyrA and parC, which alter DNA gyrase and topoisomerase IV, reducing drug binding affinity. Additionally, efflux pumps and biofilm formation can exacerbate resistance, especially in nosocomial settings [35,36]. In contrast, *Streptococcus* spp. exhibit resistance through mutations in the same target genes (gyrA and parC), with limited horizontal gene transfer [32,35]. In *Corynebacterium* spp., resistance involves a combination of target gene mutations and the acquisition of plasmid-mediated resistance genes, which shield DNA gyrase from quinolone inhibition [32]. A recent investigation of conjunctival *Corynebacterium macginleyi* isolates revealed that over 90% of fluoroquinolone-resistant strains harbor mutations in the quinolone resistance-determining region (QRDR) of the gyrA gene, predominantly at Ser-87 and Asp-91 positions [33]. Biofilm-forming strains display higher resistance, likely due to the impaired penetration of fluoroquinolones into biofilms [37,38]. These patterns mirror global reports linking systemic fluoroquinolone consumption to ocular AMR, emphasizing the interconnectedness of systemic and local antibiotic ecosystems [34,39,40,41]. While a single-center study in Andhra Pradesh similarly observed a mild decline in fluoroquinolone susceptibility among corneal bacterial isolates, this trend was not statistically significant, potentially due to variations in bacterial composition or regional antibiotic use practices [42].

In comparison, *Pseudomonas aeruginosa* retained susceptibility to aminoglycosides (amikacin: 0.88) and cephalosporins (ceftazidime: 0.96), potentially due to their distinct resistance mechanisms. The stability of *Pseudomonas aeruginosa* susceptibility to these agents may reflect the conserved nature of their targets and limited horizontal gene transfer in ocular isolates [29]. However, the post-pandemic decline in susceptibility to ofloxacin and tobramycin among Gram-negative isolates (*p* < 0.05) suggests emerging risks, potentially tied to prolonged topical antibiotic exposure in high-risk populations. This could result from efflux pump overexpression or enzymatic inactivation, emphasizing the need for vigilant stewardship even in historically effective antibiotic classes [32,35,36].

Alterations in patient behavior and healthcare access have also contributed to shaping post-pandemic trends. The widespread adoption of telemedicine and remote consultations during COVID-19 altered the management of ocular infections, potentially leading to delayed presentations and shifts in prescribing practices [16]. Patients with milder infections may have been managed conservatively without immediate antibiotic intervention, while those with more severe cases might have received empirical antibiotic therapy before culture results were available [39]. These adjustments in clinical workflows may have inadvertently influenced bacterial ecology and resistance patterns over time. Moreover, evolving clinical practices, such as revised empirical treatment protocols and a more cautious approach to antibiotic prescribing, have likely affected the selection pressure on ocular pathogens. These operational and therapeutic shifts are crucial for understanding the complex dynamics of antimicrobial resistance development in the context of a global health crisis. This study has some limitations. As a single-center analysis, regional variations in pathogen distribution and prescribing practices may limit generalizability. The retrospective design precluded assessment of clinical outcomes or detailed antibiotic exposure histories. Furthermore, the post-pandemic period (2023–2024) spans only 18 months, necessitating longer-term surveillance to confirm trends. Future investigations should prioritize molecular typing to elucidate resistance mechanisms and explore correlations between systemic antibiotic use and ocular AMR.

In summary, the COVID-19 pandemic has reshaped the microbiological landscape of BK through declining Gram-positive cocci, emerging Gram-positive bacilli, and evolving resistance to first-line antibiotics. These changes highlight the importance of adaptive antimicrobial stewardship and tailored empirical therapy. Continuous surveillance, combined with multidisciplinary efforts to curb unnecessary antibiotic use, will be critical to mitigating resistance and preserving vision in a post-pandemic era.

## 5. Conclusions

This study reveals significant changes in the bacterial pathogen profile of bacterial keratitis from 2014 to 2024, particularly during and after the COVID-19 pandemic, showing a notable reduction in Gram-positive cocci (e.g., *Staphylococcus epidermidis*) and a significant increase in Gram-positive bacilli (e.g., *Corynebacterium* spp.) following the COVID-19 pandemic; the study confirms a substantial increase in resistance to fluoroquinolone antibiotics (e.g., gatifloxacin) among BK during the COVID-19 period. These shifts, potentially linked to COVID-19-associated antibiotic overuse, highlight the need to reevaluate empirical therapies in ophthalmology and strengthen antimicrobial stewardship programs targeting both systemic and ocular antibiotic use.

## Figures and Tables

**Figure 1 microorganisms-13-00670-f001:**
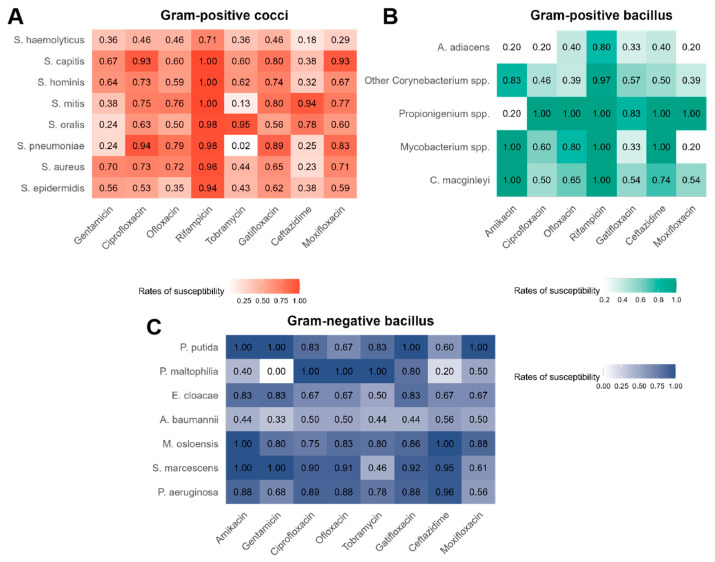
The overall susceptibility rates of bacterial isolates from cornea to various types of antibiotics among (**A**) Gram-positive cocci, (**B**) Gram-positive bacilli, and (**C**) Gram-negative bacilli. Number represents the rates of susceptibility of bacteria to antibiotics.

**Figure 2 microorganisms-13-00670-f002:**
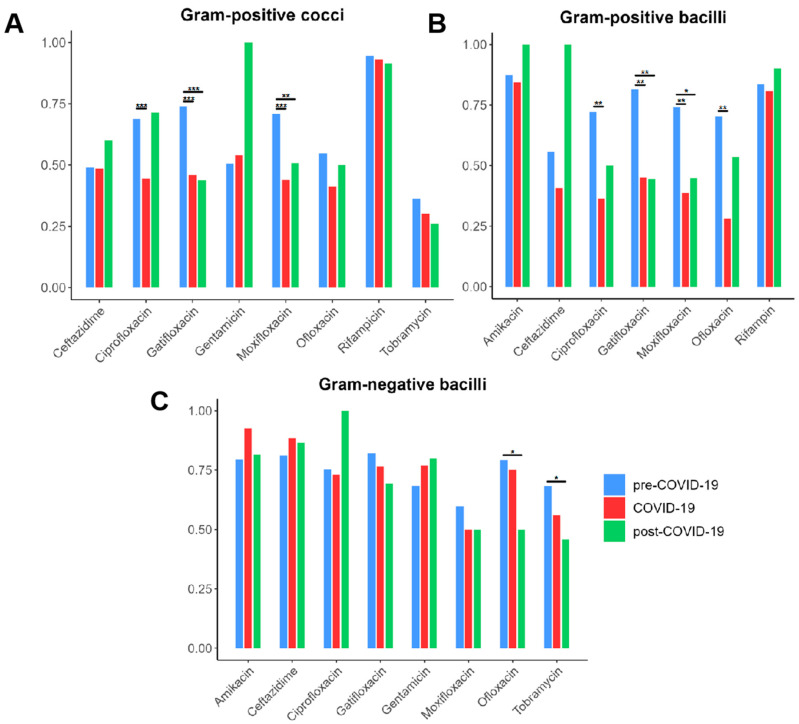
The antibiotic susceptibility rates of bacterial isolates from cornea pre-COVID-19, COVID-19, and post-COVID-19 periods among (**A**) Gram-positive cocci; (**B**) Gram-positive bacilli; (**C**) Gram-negative bacilli. * represents *p* < 0.05, ** represents 0.001 < *p* < 0.01, *** represents *p* < 0.001.

**Figure 3 microorganisms-13-00670-f003:**
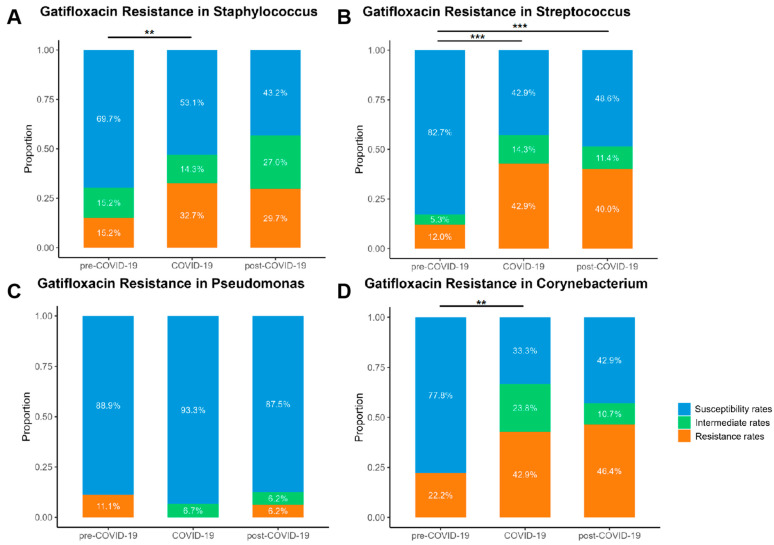
The changes in gatifloxacin resistance among (**A**) *Staphylococcus*, (**B**) *Streptococcus*, (**C**) *Pseudomonas*, and (**D**) *Corynebacterium* across the pre-COVID-19, COVID-19, and post-COVID-19 periods. Susceptibility rates, Intermediate rates, and Resistance rates mean the rates of susceptibility, intermediate, and resistance of bacteria to gatifloxacin. ** represents 0.001 < *p* < 0.01, *** represents *p* < 0.001. The total does not reach 100% due to rounding.

**Table 1 microorganisms-13-00670-t001:** Comparison analysis of bacterial isolates from the cornea among the periods of pre-COVID-19, COVID-19, and post-COVID-19.

Bacteria	Total N (%)	Pre-COVID-19 N (%)	COVID-19 N (%)	Post-COVID-19 N (%)	*p* Value
Gram-positive cocci	674 (62.9%)	460 (69.8%)	110 (54.7%)	104 (49.3%)	**<0.001 ***
*Staphylococcus epidermidis*	285 (26.6%)	205 (31.1%)	43 (21.4%)	37 (17.5%)	**<0.001 ***
*Staphylococcus aureus*	54 (5.0%)	33 (5.0%)	10 (5.0%)	11 (5.2%)	0.992
*Streptococci* spp.	202 (18.9%)	133 (20.2%)	33 (16.4%)	36 (17.1%)	0.371
*Abiotrophia* spp.	11 (1.0%)	8 (1.2%)	2 (1%)	1 (0.5%)	0.649
*Enterococcus*	10 (0.9%)	2 (0.3%)	6 (3.0%)	2 (0.9%)	**0.003 ***
*Kocuria* spp.	7 (0.7%)	5 (0.8%)	0 (0%)	2 (0.9%)	0.424
*Micrococci*	4 (0.4%)	1 (0.2%)	0 (0%)	3 (1.4%)	**0.020 ***
Other Gram-positive cocci	101 (9.4%)	73 (11.1%)	16 (8.0%)	12 (5.7%)	0.048
Gram-positive bacilli	139 (13.0%)	57 (8.6%)	37 (18.4%)	45 (21.3%)	**<0.001 ***
*Corynebacterium* spp.	84 (7.8%)	28 (4.2%)	22 (10.9%)	34 (16.1%)	**<0.001 ***
*Nocardia* spp.	12 (1.1%)	4 (0.6%)	4 (2.0%)	4 (1.9%)	0.130
*Mycobacterium*	9 (0.8%)	2 (0.3%)	6 (3.0%)	1 (0.5%)	**0.001 ***
Other Gram-positive bacilli	34 (3.2%)	23 (3.5%)	5 (2.5%)	6 (2.8%)	0.742
Gram-negative bacilli	252 (23.5%)	139 (21.1%)	54 (26.9%)	59 (28%)	0.057
*Pseudomonas* spp.	84 (7.8%)	48 (7.3%)	18 (9.0%)	18 (8.5%)	0.681
*Klebsiella* spp.	21 (2%)	14 (2.1%)	3 (1.5%)	4 (1.9%)	0.850
*Serratia* spp.	28 (2.6%)	14 (2.1%)	9 (4.5%)	5 (2.4%)	0.182
*Moraxella* spp.	25 (2.3%)	10 (1.5%)	6 (3.0%)	9 (4.3%)	0.056
*Acinetobacter* spp.	17 (1.6%)	12 (1.8%)	1 (0.5%)	4 (1.9%)	0.389
*Haemophilus* spp.	8 (0.7%)	4 (0.6%)	1 (0.5%)	3 (1.4%)	0.441
*Enterobacter* spp.	8 (0.7%)	5 (0.8%)	3 (1.5%)	0 (0%)	0.213
Other Gram-negative bacilli	61 (5.7%)	32 (4.9%)	13 (6.5%)	16 (7.6%)	0.288
Gram-negative cocci	6 (0.6%)	3 (0.5%)	0 (0%)	3 (1.4%)	0.130
Total	1071 (100%)	659 (100%)	201 (100%)	211 (100%)	

Note: Bacterial groups with significant differences (*p* < 0.05) across the periods are marked with *, and the corresponding *p*-values are displayed in bold.

## Data Availability

The information used in our study is available upon request from the corresponding author. The dataset is not available to the public due to the need to protect patient privacy.

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
