# Peer review of "Epidemiological and Antimicrobial Resistance Trends in Bacterial Keratitis: A Hospital-Based 10-Year Study (2014–2024)"

_microorganisms, 2025, doi:10.3390/microorganisms13030670_

Round 1
Reviewer 1 Report
Comments and Suggestions for Authors
The study presents a valuable retrospective analysis of bacterial keratitis over a decade, with a novel focus on the impact of the COVID-19 pandemic on pathogen distribution and antimicrobial resistance trends.
The introduction provides a comprehensive background, but a clearer explanation of the rationale behind dividing the study into pre-, during-, and post-pandemic periods would improve coherence.
The methods are well-described and follow standard microbiological and statistical practices. However, additional details on how potential confounding factors were addressed would strengthen the study’s reliability.
The results are well-structured and supported by statistical analysis, but some figures and tables could benefit from clearer legends and formatting to enhance readability.
The discussion effectively interprets the findings, but some claims regarding resistance trends could be better supported by referencing additional studies on antimicrobial resistance in ophthalmology.
The language is generally clear, but minor grammatical and stylistic refinements would improve readability. Some sentences are complex and could be simplified for clarity.
The study’s conclusions are well-aligned with the findings, highlighting important trends in bacterial epidemiology and resistance. However, a more explicit discussion on the potential clinical implications of the findings would be beneficial.
Minor comments:
Abbreviations should be sorted from A to Z.
AHYAN ILMAN QUDSI -->Why do you use capital letters for this one author?
Comments on the Quality of English LanguageThe English language in the manuscript is generally clear and understandable, but there are areas where improvements could enhance readability and clarity. Some sentences are complex and could be simplified for better flow. Minor grammatical errors and awkward phrasing are present in some sections, particularly in the discussion. Refining sentence structure and improving coherence between ideas would strengthen the overall presentation. Consider professional language editing to enhance fluency and readability.
Examples found:
Original sentence:
"Bacterial keratitis (BK) is a vision-threatening emergency, causing rapid corneal destruction and permanent vision loss, with rising antimicrobial resistance (AMR) exacerbating treatment challenges."
Suggested revision:
"Bacterial keratitis (BK) is a serious condition that can lead to rapid corneal destruction and permanent vision loss. The increasing problem of antimicrobial resistance (AMR) further complicates treatment."
Reason: The original sentence is long and contains multiple clauses, making it slightly difficult to read. Splitting it into two sentences improves clarity.
Original sentence:
"Several studies have reported changes on the prevalence and AMR profiles of bacterial pathogens in different infections during and after the COVID-19 pandemic."
Suggested revision:
"Several studies have reported changes in the prevalence and antimicrobial resistance (AMR) patterns of bacterial pathogens during and after the COVID-19 pandemic."
Reason: "Changes on" is incorrect; the correct phrase is "changes in." Also, "profiles" is slightly vague; "patterns" fits better in this context.
Original sentence:
"The results showed significant shifts in pathogen distribution, with a notable decline in Gram-positive cocci (from 69.8% pre-COVID-19 to 49.3% in post-COVID-19, P < 0.001), particularly Staphylococcus epidermidis."
Suggested revision:
"The results indicate significant changes in pathogen distribution, including a marked decrease in Gram-positive cocci (from 69.8% pre-COVID-19 to 49.3% post-COVID-19, P < 0.001), particularly in Staphylococcus epidermidis."
Reason: "Showed" is a weaker verb compared to "indicate," and "marked decrease" flows better than "notable decline."
Original sentence:
"This decline was particularly pronounced for Staphylococcus epidermidis, which decreased from 31.1% (205/659) pre-COVID-19 to 17.5% (37/211) post-COVID-19 (P < 0.001)."
Suggested revision:
"This decline was especially noticeable in Staphylococcus epidermidis, which fell from 31.1% (205/659) pre-COVID-19 to 17.5% (37/211) post-COVID-19 (P < 0.001)."
Reason: "Particularly pronounced" is somewhat formal and can be replaced with "especially noticeable" for improved readability.
Original sentence:
"These findings highlight the rapid emergence of fluoroquinolone resistance in multiple bacterial groups, underscoring the urgent need for ongoing surveillance and optimized antimicrobial stewardship to improve treatment outcomes."
Suggested revision:
"These findings emphasize the rapid rise of fluoroquinolone resistance in several bacterial groups, highlighting the need for continuous surveillance and improved antimicrobial stewardship to enhance treatment outcomes."
Reason: "Highlight" and "underscore" are somewhat redundant. "Rapid emergence" can be more clearly stated as "rapid rise," and "continuous" is more natural than "ongoing" in this context.
Original sentence:
"The study confirms a significant rise in resistance to fluoroquinolone antibiotics (such as gatifloxacin) among BK during the COVID-19 period, suggesting the need to optimize empirical antimicrobial treatment strategies."
Suggested revision:
"The study confirms a substantial increase in fluoroquinolone resistance (e.g., gatifloxacin) in bacterial keratitis during the COVID-19 period, emphasizing the need to refine empirical antimicrobial treatment strategies."
Reason: "Significant rise" is slightly vague; "substantial increase" is more precise. "Suggesting the need" is weaker than "emphasizing the need."
Original sentence:
"Empirical overuse of systemic fluoroquinolones for respiratory infections—despite their lack of efficacy against SARS-CoV-2—likely exerted cross-selective pressure on ocular pathogens."
Suggested revision:
"The widespread use of systemic fluoroquinolones for respiratory infections—despite their ineffectiveness against SARS-CoV-2—may have contributed to increased selective pressure on ocular pathogens."
Reason: "Empirical overuse" is slightly unclear; "widespread use" is more natural. "Exerted cross-selective pressure" is technical, and "may have contributed to increased selective pressure" is easier to understand.
Reviewer 2 Report
Comments and Suggestions for Authors
Estimated Authors,
I've read with great interest the present paper entitled "Epidemiological and Antimicrobial Resistance Trends in Bacterial Keratitis: A Hospital-Based 10-Year Study (2014–2024)".
Through a single center study design, Authors were able to document some interesting and significant trends on the AMR of pathogens isolated from cases of bacterial keratitis. The study, including a total of 1071 patients, documented (the following summary is quoted from the abstract of the study):
"a notable decline in Gram-positive cocci (from 69.8% pre-COVID-19 to 49.3% in post-COVID-19, P < 0.001), particularly Staphylococcus epidermidis. In contrast, Gram-positive bacilli, particularly Corynebacterium spp., increased from 4.2% to 16.1% (P < 0.001). The susceptibility to gatifloxacin, moxifloxacin, and ciprofloxacin significantly declined in both Gram-positive cocci and bacilli during the COVID-19 period (all P < 0.01). Gatifloxacin resistance in Staphylococcus rose from pre-COVID-19 (15.2%) to COVID-19 (32.7%), remaining high post-COVID-19 (29.7%). A similar trend was observed in Streptococcus and Corynebacterium, where resistance rose 24 sharply from 12.0% and 22.2% pre-COVID-19 to 42.9% during COVID-19, and remained elevated at 40.0% and 46.4% post-COVID-19, respectively (P < 0.01)".
These results, as repetitively stressed by Authors across the main text, are both consistent with our understanding of this specific theme and characterized by some specifities, accurately discussed.
Authors do their best in providing a descriptive analysis of this specific theme, not providing a potential analysis of risk factors associated with the AMR status of isolates. On the other hand, they do not try to over-discuss or over-analyze their data, and therefore I've no specific requests on data analysis and reworking of main text.
Still, some minor improvements could be suggested in terms of reporting.
First, please change all iterations of p = 0.000 to p < 0.001 Second, provide some information about the center where the study was performed (i.e. whether it was a primary or secondary center; how many patients were usually treated, and so on; of particular interest would be providing data on the total number of patients followed during the COVID-19 and immediately thereafter). Third, please note that Discussion section starts with some sentences that should be removed (201-203). Fourth, rows 211-213 contains repetitive recall of reference 15 that is similarly referenced in the introduction. Please simplify.Author Response
Please see the attachment.

Round 2
Reviewer 1 Report
Comments and Suggestions for Authors
Thank you for addressing all reviewer comments. The manuscript has been revised satisfactorily and now meets the necessary standards for publication. The study presents valuable findings on the epidemiological and antimicrobial resistance trends in bacterial keratitis, and the analysis is well-structured and informative. No further revisions are required. Congratulations on your work!